# Resolution of Fetal Hydrops Dependent on Sustained Fetal Supraventricular Tachycardia after Digoxin Therapy

**DOI:** 10.3390/medicina56050223

**Published:** 2020-05-07

**Authors:** Aureja Maciuleviciute, Migle Semenaite, Vladas Gintautas, Regina Maciuleviciene, Aras Puodziukynas, Egle Savukyne

**Affiliations:** 1Department of Obstetrics and Gynecology, Lithuanian University of Health Sciences, LT-50161 Kaunas, Lithuania; migle.semenaite@stud.lsmu.lt (M.S.); vladas.gintautas@lsmuni.lt (V.G.); regina.maciuleviciene@lsmu.lt (R.M.); egle.savukyne@lsmu.lt (E.S.); 2Department of Cardiology, Lithuanian University of Health Sciences, LT-50161 Kaunas, Lithuania; aras.puodziukynas@lsmu.lt

**Keywords:** fetal tachyarrhytmia, fetal hydrops, digoxin, sotalol

## Abstract

We present a special case of fetal supraventricular tachycardia detected at 34 weeks gestation. Fetal hydrops was noted on ultrasound upon admission. Normal fetal heart rate was maintained for three weeks by maternal administration of digoxin. A live infant was delivered via caesarian section at 37 weeks gestation. This clinical case demonstrated that pharmacological treatment can be effective and helps to prolong pregnancy safely.

## 1. Introduction

A fetal heart rate (FHR) between 110 and 160 bpm is considered to be normal. Fetal arrhythmias are irregular cardiac rhythms and/or heart rates beyond the normal range; they account for 2% of unselected pregnancies and for 10–20% of referrals to fetal cardiologists [1,2]. Fetal supraventricular tachycardia (SVT) is described as 1:1 atrioventricular activity with heart rates most commonly ranging from 200 to 300 bpm [3]. It is the most common type of fetal tachyarrhythmia (60–80%), and the prevalence ranges from 1/1000 to 1/25,000 pregnancies [4]. If untreated, sustained SVT causes elevated central venous pressure and reduced cardiac output. This may lead to fetal hemodynamic compromise and result in the development of nonimmune fetal hydrops (FH) that can cause fetal death. Multiple studies have found digoxin, flecainide, sotalol, amiodarone and other antiarrhythmic agents to be effective in the treatment of fetal SVT [5]. In the presence of FH, digoxin is poorly transferred to the fetus and can be ineffective; however, it is mostly described as the first-line agent in cases of fetal SVT [6]. We present a case report in which maternally administered digoxin successfully restored a normal FHR and reversed FH.

## 2. Case Report

A 34-year-old gravida 3, para 1, aborta 1, noted a loss of fetal motion at 34 weeks of gestation. A two-dimensional ultrasound showed fetal supraventricular tachycardia (fetal heart rate: 284 bpm), signs of fetal hydrops—ascites, pleural and pericardial effusion (Figure 1 and Figure 2). No cardiac or other structural fetal anomalies were found, as well as no anemia—middle cerebral arterypeak systolic velocity was 29 cm/s. We report on a male fetus, biometrical data adequate to gestational age. Electronic fetal heart rate monitoring failed to track the FHR due to its rapidity. The patient had a history of laboratory confirmed influenza at 30 weeks of gestation and was treated with oseltamivir. She gave birth to a healthy boy five years ago.

She was administered a single dose of propafenone 140 mg intravenously (IV), as the treatment was aimed at lowering conduction velocity of atrioventricular node and possible accessory pathways. The treatment was initiated with propafenone instead of digoxin, with the expectation of an earlier onset of clinical effect as fetal condition was deteriorating (considering the loading period of digoxin). Conversion to a normal FHR was achieved after the injection. However, supraventricular tachycardia recurred after several hours. Despite administration of oral propafenone 150 mg and metoprolol 50 mg, fetal arrhythmia persisted.

On the next day, loading with digoxin was initiated starting with 0.25 mg dose IV which was repeated after 2.5 h as the fetal arrhythmia persisted. However, FHR remained abnormal, but the patient refused the third dose of digoxin and desired to continue pregnancy with only intensive monitoring of fetal condition. Twelve hours later, fetal tachycardia resolved. Dexamethasone 12 mg was initiated for fetal lung maturation, followed by oral digoxin therapy of 0.25 mg daily. Daily FHR monitoring (four times per day) continued, and ultrasound scans were performed twice every day until 36 weeks of gestation. Electrolytes along with digoxin levels were checked every second day, and serum therapeutic levels between 0.5 and 1.2 nmol/L were maintained. The fetal ascites and pleural and pericardial effusion appeared to be resolving by the third day in hospital. The patient was discharged on the same digoxin dosage with close outpatient follow-ups at 36 weeks of gestation.

The patient was admitted to the labor and delivery unit at 37 weeks of gestation due to loss of fetal movements. Fetal supraventricular tachycardia was found again at that moment without signs of FH. Laboratory tests revealed digoxin level of 0.44 ng/mL (therapeutic range: 0.5–1.2 nmol/L), a potassium level of 3.6 mmol/L (normal range: 3.6–5.1 mmol/L), magnesium level of 0.7 mg/dL (normal range: 0.74–1.03 mmol/L), and ionized calcium level of 0.98 mg/dL (normal range: 1.2–1.43 mmol/L). Oral sotalol 80 mg was administered every eight hours. After two doses, fetal tachyarrhythmia persisted, therefore intravenous digoxin 0.25 mg was added. However, no effect was achieved. Furthermore, signs of heart failure (ascites) were noted on ultrasound. As there was no possibility for a rapid natural delivery, emergency cesarean section was done (Appendix A, Figure A1).

A live infant weighing 3612 g, Apgar scores of 7 (1 min) and 8 (5 min), and umbilical cord blood hemoglobin concentration of 175 g/L was born. Tachyarrhythmia of 200 bpm, tachypnea, peripheral cyanosis, and edema were observed. The blood pressure was unmeasurable. In preparation for cardioversion, the newborn was intubated, which resulted in normal heart rate (possibly acting as vagal stimulation maneuver). However, tachycardia recurred after six hours, adenosine was administered with no effect, and the newborn underwent cardioversion with successful conversion to sinus rhythm and a heart rate of 130 bpm. Based on surface electrocardiogram (ECG), Wolff–Parkinson–White syndrome and paroxysmal supraventricular tachycardia were diagnosed. Later, the newborn showed no recurrence of tachyarrhythmia, as determined by ECG monitoring during the period of hospitalization, and was discharged from the hospital nine days after birth. The parents were advised to measure the pulse regularly and continue follow-up in the pediatric cardiology outpatient department. In the first month of life, the infant was developing well with normal weight gain, and there was no recurrence of tachyarrhythmia.

## 3. Discussion

Our clinical case demonstrated successful conversion and control of fetal SVT using digoxin. Even though the fetal heart was affected, the main mode of treatment was transplacental, by administering the drug to the mother. The aim of in utero therapy is to prevent or reverse heart failure by managing the tachyarrhythmia. More than 80% of fetal tachycardias can be treated successfully; but for some tachyarrhythmias, particularly in the presence of hydrops, more than one medication and several days of therapy may be necessary [2]. Therefore, urgent caesarean section should be reserved in case of deterioration of fetal condition despite adequate treatment.

According to the American Heart Association, treatment of fetal SVT should be based on maternal condition, gestational age, intermittent versus sustained tachycardia, degree of tachycardia, presence and degree of fetal compromise, and whether hydrops is present or not [7]. Digoxin has traditionally been the first-line drug in the management of SVT in utero. It prolongs the refractory period in the atrioventricular (AV) node, which results in suppressed conduction. In addition to primary electrophysiological properties, it also has possible inotropic, sympathomimetic, and antiarrhythmic properties [4]. In hospitals, maternal, fetal, and plasma drug level monitoring is required to avoid fetal and maternal toxicity at initiation of therapy. Timely recognition of the signs, symptoms, and laboratory test results help to prevent complications of digoxin overdose. Clinical symptoms of digoxin toxicity include multiple arrhythmias (AV blocks, atrial and ventricular fibrillation, etc). Renal dysfunction and electrolyte disbalance such as hypokalemia, hypercalcemia, and hypomagnesemia can precipitate digoxin toxicity. Extracardiac side effects include abdominal pain, nausea, vomiting, weakness, and confusion [8].

Alternatively, many authors suggest sotalol, alone or in combination with digoxin, instead of digoxin monotherapy, especially in cases of FH [8,9]. In 2012, Shah et al. demonstrated an 85% complete or partial response rate to treatment with transplacental sotalol [10]. Although sotalol is generally well tolerated, cardiac rhythm monitoring and assessment should be performed when sotalol therapy is initiated or when doses are increased due to its proarrhythmic effect [11].

In 2017, Alsaied et al. reported the dosing and route of administration of the medications in a systematic review and meta-analysis of literature on treatment of fetal tachyarrhythmia [5]. The recommended starting dose of Digoxin is loading with 1.5–2 mg over 24–48 h, oral (PO)or IV (maintenance 0.375–1 mg/day targeting maternal drug levels 1–2.5 ng/mL). The starting dose of Flecainide is 200–300 mg divided two times a day (BID) or three times a day (TID) PO The dose can be increased to 450 mg/day if no response is reached (maternal drug levels of 0.2–1 L/mL are targeted). The authors recommend starting sotalol with 160–320 mg divided BID PO (maximum dose 480 mg/day), while Amiodarone should be started with a loading dose of 1600–2400 mg/day two to four times daily PO or IV. The dose is usually halved every 24 h until the maintenance dose of 200–400 mg/day BID is reached.

Propafenone used in our case has the same electrophysiologic mechanism of action as flecainide. Amiodarone is generally avoided due to multiple extracardiac side effects.

## 4. Concluding Remarks

Despite the lack of agreement on the most effective and best tolerated first-line medication for the treatment of fetal SVT, pharmacological treatment should be administered to all pregnant women with sustained fetal tachycardia. Pregnancy termination should only be considered if there is no response to treatment. It is a rare condition, and management could be challenging as the doctor is required to weigh up the risk of early delivery with potential preterm complications, against the safety and effectiveness of available therapies. Each case is different and requires individualized care.

## Figures and Tables

**Figure 1 medicina-56-00223-f001:**
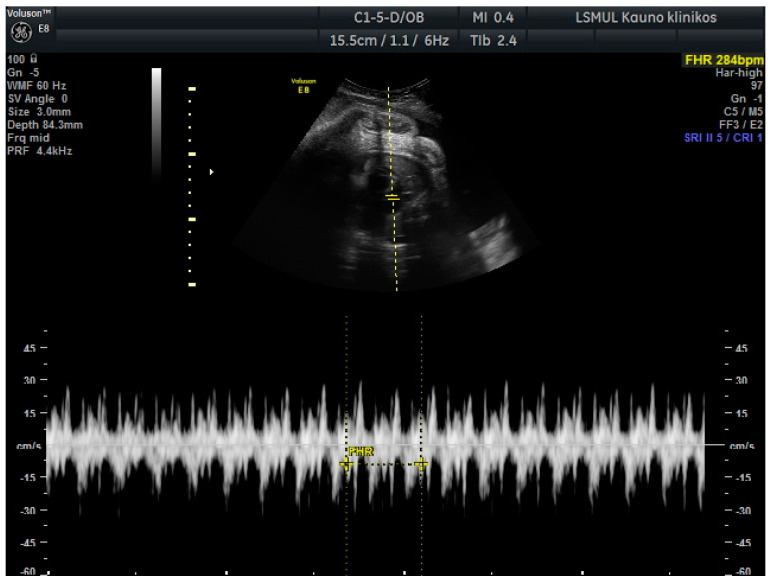
Fetal supraventricular tachycardia.

**Figure 2 medicina-56-00223-f002:**
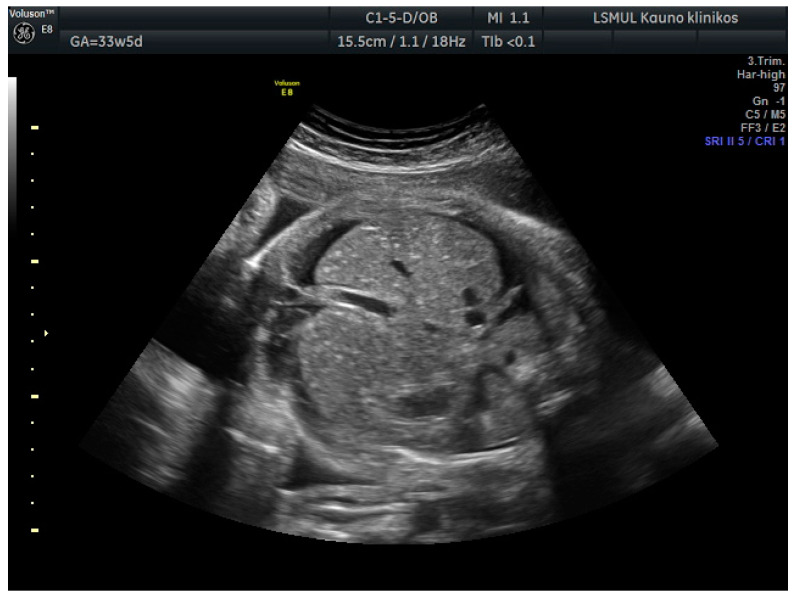
Ascites.

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
