# Peer review of "Resolution of Fetal Hydrops Dependent on Sustained Fetal Supraventricular Tachycardia after Digoxin Therapy"

_medicina, 2020, doi:10.3390/medicina56050223_

Round 1
Reviewer 1 Report
The manuscript entitled “Resolution of fetal hydrops dependent on sustained fetal supraventricular tachycardia after digoxin therapy” presents a case of a drug reversed fetal supraventricular tachycardia. The topic of this Manuscript is interesting and falls within the scope of Medicina. However, the abstract section should be improved. The Case report section needs some changes as reported below. The study needs major revisions as detailed in the “Reviewer Blind Comments to Author”.
- All the text needs a language revision by a native English speaker person, in order to correct typos and style (repetitions, punctuation, grammar):
- Case report: was this study designed according to CARE guidelines (PMID: 24228906), available through the EQUATOR Network (http://www.equatornetwork.org/)? It would be mandatory to report this information in the text.
- Case report: Why propafenone was the first choice in the treatment? You should underline this point.
- Case report: You should consider using a flow diagram to better show the episodes of care of the patient.
Thanks
Author Response
Point 1: Case report: was this study designed according to CARE guidelines (PMID: 24228906), available through the EQUATOR Network (http://www.equatornetwork.org/)? It would be mandatory to report this information in the text.
Response 1: We have not followed CARE guidelines preparing our case report directly, although majority of aspects of the clinical case, recommended in CARE guidelines were covered.
Point 2: Why propafenone was the first choice in the treatment? You should underline this point.
Response 2: The treatment was initiated with propafenone instead of digoxin expecting earlier onset of clinical effect as fetal condition was deteriorating (considering the loading period of digoxin).
Point 3: You should consider using a flow diagram to better show the episodes of care of the patient.
Response 3: We added a flow diagram in our clinical case as Appendix 1.
We have improved the abstract section as advised and had the language reviewed by a native English speaker person.
Reviewer 2 Report
Overall well written case report with good review of available literature
Author Response
Thank you for your review.
Round 2
Reviewer 1 Report
Authors followed the reviewers' instructions. Thank You
This manuscript is a resubmission of an earlier submission. The following is a list of the peer review reports and author responses from that submission.